# Implicitly Learning to Reason in First-Order Logic

**Vaishak Belle**
University of Edinburgh & Alan Turing Institute
`vaishak@ed.ac.uk`

**Brendan Juba**
Washington University in St. Louis
`bjuba@wustl.edu`

## Abstract

We consider the problem of answering queries about formulas of first-order logic based on background knowledge partially represented explicitly as other formulas, and partially represented as examples independently drawn from a fixed probability distribution. PAC semantics, introduced by Valiant, is one rigorous, general proposal for learning to reason in formal languages: although weaker than classical entailment, it allows for a powerful model theoretic framework for answering queries while requiring minimal assumptions about the form of the distribution in question. To date, however, the most significant limitation of that approach, and more generally most machine learning approaches with robustness guarantees, is that the logical language is ultimately essentially propositional, with finitely many atoms. Indeed, the theoretical findings on the learning of relational theories in such generality have been resoundingly negative. This is despite the fact that first-order logic is widely argued to be most appropriate for representing human knowledge. In this work, we present a new theoretical approach to robustly learning to reason in first-order logic, and consider universally quantified clauses over a countably infinite domain. Our results exploit symmetries exhibited by constants in the language, and generalize the notion of implicit learnability to show how queries can be computed against (implicitly) learned first-order background knowledge.

## 1   Introduction

The tension between *deduction* and *induction* is perhaps the most fundamental issue in areas such as philosophy, cognition and artificial intelligence. The deduction camp concerns itself with questions about the expressiveness of formal languages for capturing knowledge about the world, together with proof systems for reasoning from such *knowledge bases*. The learning camp attempts to generalize from examples about partial descriptions about the world. In an influential paper, Valiant [31] recognized that the challenge of learning should be integrated with deduction. In particular, he proposed a semantics to capture the quality possessed by the output of (probably approximately correct) PAC-learning algorithms when formulated in a logic. Although weaker than classical entailment, it allows for a powerful model theoretic framework for answering queries.

From the standpoint of learning an expressive logical knowledge base and reasoning with it, most PAC results are somewhat discouraging. For example, in agnostic learning [12] where one does not require examples (drawn from an arbitrary distribution) to be fully consistent with learned sentences, efficient algorithms for learning conjunctions would yield an efficient algorithm for PAC-learning DNF (also over arbitrary distributions), which current evidence suggests to be intractable [6]. Thus, it is not surprising that when it comes to first-order logic (FOL), very little work tackles the problem in a general manner. This is despite the fact that FOL is widely argued to be most appropriate for representing human knowledge (e.g., [23, 26, 18]). For example, [4] consider the problem of the learnability of description logics with equality constraints. While description logics are already restricted fragments of FOL in only allowing unary and some binary predicates, it is shown that such a fragment cannot be tractably learned, leading to the identification of syntactic restrictions for

learning from positive examples alone. Analogously, when it comes to the learning of logic programs [5], which in principle may admit infinitely many terms, syntactic restrictions are also typical [7].

In this work, we present new results on learning to reason in FOL knowledge bases. In particular, we consider the problem of answering queries about FOL formulas based on background knowledge partially represented explicitly as other formulas, and partially represented as examples independently drawn from a fixed probability distribution. Our results are based on a surprising observation made in [11] about the advantages of eschewing the explicit construction of a hypothesis, leading to a paradigm of *implicit learnability*. Not only does it enable a form of agnostic learning while circumventing known barriers, it also avoids the design of an often restrictive and artificial choice for representing hypotheses. (See, for example, [14], which is similar in spirit in allowing declarative background knowledge but only permits constant-width clauses.) In particular, implicit learning allows such learning from partially observed examples, which is commonplace when knowledge bases and/or queries address entities and relations not observed in the data used for learning.

That work was limited to the propositional setting, however. Here, we develop a first-order logical generalization. This requires us to generalize the notions of validity and entailment, and propose new methods for recognizing true formulas under partial information, that capture what is implicitly learned. Since reasoning in full FOL is undecidable we need to consider a fragment, but the fragment we identify and are able to learn and reason with is expressive and powerful. Consider that standard databases correspond to a maximally consistent and finite set of literals: every relevant atom is known to be true and stored in the database, or known to be false, inferred by (say) negation as failure. Our fragment corresponds to a consistent but infinite set of ground clauses, not necessarily maximal. To achieve the generalization, we revisit the PAC semantics and exploit symmetries exhibited by constants in the language. Moreover, the underlying language is general in the sense that no restrictions are posed on clause length, predicate arity, and other similar technical devices seen in PAC results. We hope the simplicity of the framework is appealing to the readers and hope our results will renew interest in learnability for expressive languages with quantificational power.

We remark that our sole focus is in PAC-semantics approaches, but there are also other families of methods for unifying statistical and logical representations, that fall under the banner of *statistical relational learning* (SRL) (e.g., [13]). SRL includes widely used formalisms such as Markov Logic Networks [28] and frameworks such as Inductive Logic Programming [27]. Learning strategies for SRL is an active area of research with numerous recent advances—for example, a family of recent works have adapted the techniques for training neural networks into the Inductive Logic Programming paradigm [3, 29, 8, 22]. Generally speaking, there are significant differences to PAC-semantics approaches, such as in terms of the learning regime, the notion of correctness, and the underlying algorithmic machinery. For example, Markov Logic Networks use approximate maximum-likelihood learning strategies to capture the distribution of the data, whereas in PAC formulations, one considers an arbitrary unknown distribution over the data and studies the question of what formulas are learnable whilst costing for the number of examples needed to be sampled from that distribution. PAC-semantics is distinguished in being able to provide guarantees of generalization performance and polynomial time complexity with the minimal assumption of i.i.d. training examples. Of course, there is much to be gained by attempting to integrate these communities; see, for example, [5]. These differences notwithstanding, the learning of logical theories is usually restricted to finite-domain first-order logic, and so it is essentially propositional, and in that regard, our setting is significantly more challenging.

## 2  Logical Framework

**Language:** We let $\mathcal{L}$ be a first-order language with equality and relational symbols $\{P(x), \ldots, Q(x_1, \ldots, x_k), \ldots\}$, variables $\{x, y, z, \ldots\}$, and a countably infinite set of *rigid designators* or *names*, say, the set of natural numbers $\mathbb{N}$, serving as the domain of discourse for quantification. Well-defined formulas are constructed using logical connectives $\{\neg, \vee, \forall, \wedge, \exists, \supset\}$, as usual. ($\supset$ denotes implication.) Together with equality, names essentially realize an infinitary version of the unique-name assumption.[1]

In general, the unique-name assumption does not rule out capturing uncertainty about the identity of objects; see [9, 30], for example.

The set of (ground) atoms is obtained as:[2] $\text{ATOMS} = \{P(a_1, \ldots, a_k) \mid P \text{ is a predicate, } a_i \in \mathbb{N}\}$. We sometimes refer to elements of ATOMS as propositions, and ground formulas as propositional formulas. We will use $p, q, e$ to denote atoms, and $\alpha, \beta, \phi, \psi$ to denote ground formulas.

**Semantics:** A $\mathcal{L}$-model $M$ is a $\{0, 1\}$ assignment to the elements of ATOMS. Using $\models$ to denote satisfaction, the semantics for $\phi \in \mathcal{L}$ is defined as usual inductively, but with equality as identity: $M \models (a = b)$ iff $a$ and $b$ are the same names, and quantification understood substitutionally over all names in $\mathbb{N}$: $M \models \forall x \phi(x)$ iff $M \models \phi(a)$ for all $a \in \mathbb{N}$. We say that $\phi$ is *valid* iff for every $\mathcal{L}$-model $M$, $M \models \phi$. Let the set of all models be $\mathcal{M}$.

**Representation:** Like in standard FOL, reasoning over the full fragment of $\mathcal{L}$ is undecidable. Interestingly, owing to a fixed, albeit countably infinite, domain of discourse, the *compactness* property that holds for classical first-order logic *does not hold* in general [17]. For example, $\{\exists x P(x), \neg P(1), \neg P(2), \ldots\}$ is an unsatisfiable theory for which every finite subset is indeed satisfiable. However, as identified in [1], and earlier in [15], the case of disjunctive knowledge is more manageable. In particular, we will be interested in learning and reasoning with incomplete knowledge bases with disjunctive information [1]:

**Definition 1:** An *acceptable* equality is of the form $x = a$, where $x$ is any variable and $a$ any name. Let $e$ range over formulas built from acceptable equalities and connectives $\{\neg, \vee, \wedge\}$. Let $c$ range over quantifier-free disjunctions of (possibly non-ground) atoms. Let $\forall \phi$ mean the universal closure of $\phi$, i.e., with a universal quantifier on each free variable of $\phi$. A formula of the form $\forall(e \supset c)$ is called a $\forall$-clause. A knowledge base (KB) $\Delta$ is proper[+] if it is a finite non-empty set of $\forall$-clauses. The *rank* of $\Delta$ is the maximum number of variables mentioned in any $\forall$-clause in $\Delta$.

This fragment is very expressive. Consider that standard databases correspond to a maximally consistent and finite set of literals, in the sense that every relevant atom is known to be true and stored in the database, or known to be false, inferred by (say) negation as failure. In contrast, proper[+] KBs correspond to a consistent but infinite set of ground clauses, not necessarily maximal in this way. We also note that [19] shows how to represent a certain family of "local" action models for planning within the fragment of proper[+] we consider, for which polynomial-time reasoning is possible.

**Grounding:** A ground theory is obtained from $\Delta$ by substituting variables with names. Suppose $\theta$ denotes a substitution. We denote the result of applying $\theta$ to a formula $\phi$ by $\phi\theta$. For any set of names $C \subseteq \mathbb{N}$, we write $\theta \in C$ to mean substitutions are only allowed wrt the names in $C$. Formally, we define:

- $\text{GND}(\Delta) = \{c\theta \mid \forall(e \supset c) \in \Delta, \theta \in \mathbb{N} \text{ and } \models e\theta\}$;
- For $z \geq 0$, $\text{GND}(\Delta, z) = \{c\theta \mid \forall(e \supset c) \in \Delta, \models e\theta, \theta \in Z\}$, where $Z$ is the set of names mentioned in $\Delta$ plus $z$ (arbitrary) new ones;
- For $C \subseteq \mathbb{N}$, $\text{GND}(\Delta, C) = \{c\theta \mid \forall(e \supset c) \in \Delta, \models e\theta, \theta \in Z\}$ where $Z$ is the set of names mentioned in $\Delta$ plus the names in $C$;
- $\text{GND}^-(\Delta) = \text{GND}(\Delta, z)$ where $z$ is the rank of $\Delta$.

**Reasoning:** Unfortunately, arbitrary reasoning with such KBs is also undecidable [15, Theorem 7]. Various proposals have appeared to consider that problem: in [15], for example, a sound but incomplete evaluation-based semantics is studied. In [1], it is instead shown that when the query is limited to ground formulas, we can reduce first-order entailment to propositional satisfiability:

**Theorem 2:** *[1] Suppose $\Delta$ is a proper[+] KB, and $\alpha$ is a ground formula. Then, $\Delta \models \alpha$ iff $\text{GND}^-(\Delta \wedge \neg\alpha)$ is unsatisfiable.*

Here, the RHS of the iff is a propositional formula, obtained by a finite grounding, as defined above.

**Example 3:** Suppose $\Delta = \{\forall x(Grad(x) \vee Prof(x)), \forall x(x \neq charles \supset Grad(x))\}$ and the query is $Grad(logan)$. The query can be seen to be entailed. Given that the KB's rank is 1, consider the grounding of the KB and the negated query wrt $\{charles, logan, jean\}$ (here *jean* is chosen arbitrarily). It is indeed unsatisfiable.

It is worth noting that the proof here (and in other proposals with $\mathcal{L}$-like languages [18, 15, 21]) is established by setting up a bijection between names to show that all names other than those that

appear in the finite grounding in the RHS behave "identically," and so for entailment purposes, it suffices to consider a finite set consisting of the constants already mentioned and a few extra ones. That idea can be traced back to [17] (reformulated here for our purposes):

**Theorem 4:** *[17] Suppose $\alpha = \forall x \phi(x)$ is a $\forall$-clause. (Its rank is 1.) Let $C$ be the names mentioned in $\text{GND}(\alpha, 1)$. Then for every $a \in \mathbb{N}$, there is a $b \in C$ such that $\models \phi(a)$ iff $\models \phi(b)$.*

The essence of Theorem 2 is to exploit this idea to show (reformulated here for our purposes):

**Lemma 5:** *[1] Suppose $\alpha$ is as above. If $\text{GND}(\alpha, 1)$ is satisfiable, then so is $\text{GND}(\alpha, z)$ for $z \geq 1$.*

Thus, because Theorem 4 establishes that $\text{GND}(\alpha, 1)$ is satisfiable if and only if $\alpha$ is in the countably infinite domain, and Lemma 5 establishes that the introduction of extra names in $\text{GND}(\alpha, z)$ preserves satisfiability, we obtain satisfiability under the larger, common subset of names used in $\text{GND}^-$. These observations will now lead to an appealing account for *implicit learnability* with proper$^+$ KBs.

## 3 Generalizing PAC-Semantics

We now recall the semantics we use, PAC semantics as introduced by Valiant [31]. PAC semantics was formulated to capture the quality possessed by the output of PAC-learning algorithms, when viewed as formulas in a logic. Because inductive generalization cannot be captured by deduction, it inherently requires we admit the possibility of an incorrect generalization. Thus, as compared to classical (Tarskian) semantics, the PAC semantics is necessarily weaker. In the classical propositional formulation, we suppose a propositional language with (say) $n$ propositions, yielding a model theoretic space $\{0, 1\}^n$. We suppose that we observe examples independently drawn from a distribution $D$ over $\{0, 1\}^n$. Then, suppose further that these examples enable a learning algorithm to find a formula $\phi$. We cannot expect this formula to be valid in the traditional sense, as PAC-learning does not guarantee that the rule holds for every possible binding, only that $\phi$ so produced agrees with probability $1 - \epsilon$ wrt future examples drawn from the same distribution. This motivates a weaker notion of validity:

**Definition 6:** Given a distribution $D$ over $\{0, 1\}^n$, we say that a Boolean function $F$ is $(1 - \epsilon)$-*valid* if $\text{Pr}_{x \in D}[F(x) = 1] \geq 1 - \epsilon$. If $\epsilon = 0$, we say $F$ is *perfectly valid*.

Thus far, the PAC semantics and its application to the formalization of robust logic-based learning has been limited to the propositional setting [31, 24, 11], that is, where the learning vocabulary is finitely many atoms, and the background knowledge is essentially restricted to a propositional formula.[3] Generalizing that to the FOL case has to address, among other things, what $(1 - \epsilon)$-validity means, how FOL formulas could be learned by algorithms, and finally, how entailments can be computed. That is precisely our goal for this paper.

We start by proposing an extension of the PAC semantics for the infinitary structures (generalizing assignments) constructed for $\mathcal{L}$, namely $\mathcal{M}$. For this, we will need to consider distributions on $\mathcal{M}$, which are defined as usual [2]: we take $\mathcal{M}$ to be the sample space (of elementary events), define a $\sigma$-algebra $\mathbf{M}$ to be a set of subsets of $\mathcal{M}$, which represent a collection of (not necessarily elementary) events, and a function $\text{Pr}\colon \mathbf{M} \to [0, 1]$, which is the probability measure.

We are now ready to define $(1 - \epsilon)$-validity as needed in the PAC semantics.

**Definition 7:** Given a distribution $\text{Pr}$ over $\mathcal{M}$, we say a formula $\phi \in \mathcal{L}$ is $(1 - \epsilon)$-valid iff $\text{Pr}([\![\phi]\!]) \geq 1 - \epsilon$. If $\epsilon = 0$, then we say that $\phi$ is perfectly valid. Here, $[\![\phi]\!]$ for any closed formula $\phi \in \mathcal{L}$ denotes the set $\{M \in \mathcal{M} \mid M \models \phi\}$.

In practice, the most important use of the notion of validity is to check the entailment of a formula from a knowledge base, and by extension, the reader may wonder how that carries over from classical validity. As also observed in [11] (for the propositional case), the union bound allows classical reasoning to have a natural analogue in the PAC semantics, shown below. Note that, as already mentioned, our assumption henceforth is that knowledge bases are proper$^+$, and queries are ground formulas, both in the context of reasoning as well as learning.

**Proposition 8:** *Let $\psi_1, \ldots, \psi_k$ be $\forall$-clauses such that each $\psi_i$ is $(1 - \epsilon_i)$-valid under a common distribution $D$ for some $\epsilon_i \in [0, 1]$. Suppose $\{\psi_1, \ldots, \psi_k\} \models \varphi$, for some ground formula $\varphi$. Then $\varphi$ is $(1 - \epsilon')$-valid under $D$ for $\epsilon' = \sum_i \epsilon_i$.*

## 4   Partial Observability

The learning problem of interest here is to obtain knowledge about the distribution $D$, which, of course, is not revealed directly, but in the form of a set of examples. The examples in question are models independently drawn from $D$, and we are then interested in knowing whether a query $\alpha$ is $(1 - \epsilon)$-valid. Intuitively, background knowledge $\Delta$ may be provided additionally and so the examples correspond to additional knowledge that the agent learns. This additional knowledge is never materialized in the form of $\mathcal{L}$-formulas, but is left implicit, as postulated first in [11].

When it comes to the examples themselves, however, we certainly cannot expect the examples to reveal the full nature of the world, and indeed, partial descriptions are commonplace in almost all applications [25]. In the case of $\mathcal{L}$, moreover, providing a full description may even be impossible in finite time. All of this motivates the following:

**Definition 9:** A partial model $N$ maps ATOMS to $\{1, 0, *\}$. We say $N$ is consistent with a $\mathcal{L}$-model $M$ iff for all $p \in$ ATOMS, if $N[p] \neq *$ then $N[p] = M[p]$. Let $\mathcal{N}$ be the set of all partial models.

Essentially, our knowledge of $D$ will be obtained from a set of partial models that are the examples.

**Definition 10:** A mask is a function $\theta$ that maps $\mathcal{L}$-models to partial models, with the property that for any $M \in \mathcal{M}$, $\theta(M)$ is consistent with $M$, and only a finite number of atoms are mapped to $\{0, 1\}$. A *masking process* $\Theta$ is a mask-valued random variable (i.e., a random function). We denote the distribution over partial models obtained by applying a masking process $\Theta$ to a distribution $D$ over $\mathcal{L}$-models by $\Theta(D)$.[4]

The definition of masking processes allows the hiding of entries to depend on the underlying example from $D$. Moreover, as discussed in [11] (for the propositional case), reasoning in PAC-Semantics from complete examples is trivial, whereas the hiding of all entries by a masking process means that the problem reduces to classical entailment. So, we expect examples to be of a sort that is in between these extremes. In particular, for the sake of tractable learning, we must consider formulas that can be evaluated efficiently from the partial models with high probability. This leads to a notion of *witnessing*.

**Definition 11:** We define a propositional formula $\phi \in \mathcal{L}$ to be witnessed to evaluate to true or false in a partial assignment $N$ by induction as follows:

- an atom $Q(\vec{c})$ is witnessed to be true/false iff it is true/false respectively in $N$;
- $\neg \phi$ is witnessed true/false iff $\phi$ is witnessed false/true respectively;
- $\phi \lor \psi$ is witnessed true iff either $\phi$ or $\psi$ is, and it is witnessed false iff both $\phi$ and $\psi$ are witnessed false;
- $\phi \land \psi$ is witnessed true iff both $\phi$ and $\psi$ are witnessed true, and it is witnessed false iff either $\phi$ or $\psi$ is witnessed false;
- $\phi \supset \psi$ is witnessed true iff either $\phi$ is witnessed false or $\psi$ is witnessed true, and it is witnessed false iff both $\phi$ is witnessed true and $\psi$ is witnessed false.

We define a $\forall$-clause $\forall \vec{x} \phi(\vec{x})$ to be witnessed true in a partial model $N$ for the set of names $C$ if for every binding of $\vec{x}$ to names $\vec{c} \in C$, the resulting ground clause $\phi(\vec{c})$ is witnessed true in $N$.

It is the witnessing of $\forall$-clauses that, in essence, enables the implicit learning of quantified generalizations. Let us see how that works. Intuitively, from examples $\phi(\vec{c}_1), \ldots,$ one would like to generalize to $\forall \vec{x} \phi(\vec{x})$, the latter being a statement about infinitely many objects. But what criteria would justify this generalization, outside of (say) witnessing infinitely many instances? Our result shows that, surprisingly, it suffices to get finitely many examples, so as to witness $\phi(\vec{c}_1), \ldots, \phi(\vec{c}_k)$ and yield

universally quantified sentences with high probability. This is possible because, via Theorem 2, all the names not mentioned in the KB and the query behave identically. Thus, provided we witness the grounding of $\phi$ for a sufficient but finite set of constants, we can treat the implicit KB as including $\forall$-clauses, as it yields the same judgments on our queries.

Putting it all together, formally, in any given learning epoch, let $S$ be the class of queries we are interested in asking: that is, $S$ is any finite set of ground formulas. Let $C$ then be all the names mentioned in $S$, the KB, and $z$ extra new ones chosen arbitrarily, where $z$ is at least the rank of the KB. If $z = $ KB's rank, then the rank of the implicit KB matches that of the explicit KB; otherwise, it would be higher. So the definition says that the witnessing of $\forall \vec{x} \phi(\vec{x})$ happens when $\phi(\vec{c})$ is witnessed for all $\vec{c} \in C$. We think this notion is particularly powerful, as it neither makes references to bindings from the full set of names $\mathbb{N}$ (which is infinite), nor to not observing negative instances. Note also that witnessing does not require observing all atoms: a clause is witnessed to evaluate to true if some literal appearing in it is true in the partial model. Thus, the $\forall$-clause witnessed may involve predicates not explicitly appearing in the partial model.

**Example 12:** Let $\Delta$ be the KB

$$\{\forall x \neq logan \supset Mutant(x), \forall x \neq y \supset [Mutant(x) \wedge Teammate(x, y) \supset Mutant(y)]\}.$$

Then the $\forall$-clause $\forall x \neq logan \supset [Mutant(x) \supset Teammate(x, logan)]$ is witnessed for a suitable set of names w.r.t. $\Delta$ (with rank two) in any example that mentions at least two other names (in addition to *logan*) for which the substitution into $Mutant(x) \supset Teammate(x, logan)$ is satisfied in the partial model. For instance, we may have the partial model {*Teammate(scott,logan),Teammate(jean,logan)*}, or the partial model {*Teammate(ororo,logan),Teammate(kurt,logan)*}.

Witnessed formulas correspond to the *implicit* KB. In order to capture the inferences that the implicit KB permits, we will use partial models to simplify complex formulas in the KB or query. To that end, we define:

**Definition 13:** Given a partial model $N$ and a propositional formula $\phi$, the *restriction of $\phi$ under $N$,* denoted $\phi|_N$, is recursively defined: if $\phi$ is an atom witnessed in $N$, then $\phi|_N$ is the value that $\phi$ is witnessed to evaluate to under $N$; if $\phi$ is an atom not set by $N$, then $\phi|_N = \phi$; if $\phi = \neg \psi$, then $\phi|_N = \neg(\psi|_N)$; and if $\phi = \alpha \wedge \beta$, then $\phi|_N = (\alpha|_N) \wedge (\beta|_N)$. (And analogously for Boolean connectives $\vee$ and $\supset$ .) For a partial model $N$ and set of propositional formulas $F$, we let $F|_N$ denote the set $\{\phi|_N : \phi \in F\}$.

Notice that here we do not define restrictions for quantified formulas, such as those appearning in the KB: while that is possible it is not needed, as we will be leveraging Theorem 2 for reasoning.

**Example 14 :** Consider $GND^-(\Delta)$ for the KB $\Delta$ of Example 12 using the set of names {*scott,jean,logan*}. Then the restriction of the grounding of our second rule under the partial model {*Teammate(scott,logan),Teammate(jean,logan),Teammate(scott,jean)*} is

*Mutant(scott)* $\supset$ *Mutant(logan)*, [*Mutant(logan)* $\wedge$ *Teammate(logan,scott)* $\supset$ *Mutant(scott)*],

*Mutant(jean)* $\supset$ *Mutant(logan)*, [*Mutant(logan)* $\wedge$ *Teammate(logan,jean)* $\supset$ *Mutant(jean)*],

*Mutant(scott)* $\supset$ *Mutant(jean)*, [*Mutant(jean)* $\wedge$ *Teammate(jean,scott)* $\supset$ *Mutant(scott)*].

Had the partial model also included *Teammate(logan,scott), Teammate(logan,jean),* and *Teammate(scott,jean)* we would have had the further simpler collection

*Mutant(scott)* $\supset$ *Mutant(logan)*, *Mutant(logan)* $\supset$ *Mutant(scott)*,

*Mutant(jean)* $\supset$ *Mutant(logan)*, *Mutant(logan)* $\supset$ *Mutant(jean)*,

*Mutant(scott)* $\supset$ *Mutant(jean)*, *Mutant(jean)* $\supset$ *Mutant(scott)*.

## 5 Implicit Learnability

The central motivation here is learning to reason in FOL, and as argued earlier, implicit learning circumvents the need for an explicit hypothesis, especially since hypothesis fitting is intractable, unless one severely restricts the hypothesis space. So, learning is integrated tightly into the application

using the knowledge extracted from data. Our definitions in the previous sections establish the grounds for which a first-order implicit KB can be learned from finitely many finite-size examples, but also the grounds for deciding propositional entailments of $\forall$-clauses specified explicitly – i.e., the background knowledge. (Of course, reasoning is not yet tractable, but simply decidable; we return to this point later). Overall, the learning regime is presented in Algorithm 1, and its correctness is justified in Theorem 15.

---

**Algorithm 1** Reasoning with implicit learning

---

**Input:** Partial models $N^{(1)}, N^{(2)}, \ldots, N^{(m)}$, explicit KB $\Delta$, query $\alpha$ (a ground formula), number of names $k$ at least equal to $\Delta$'s rank
**Output:** $\hat{p} \in [0, 1]$ estimating $\alpha$ is $\hat{p}$-valid (See Theorem 15)
Initialize $v \leftarrow 0$
**for** $i = 1, \ldots, m$ **do**
    **for** all $k$-tuples of names $(c_1, \ldots, c_k)$ from $N^{(i)}$ *not* appearing in $\Delta \wedge \neg\alpha$ **do**
        **if** $\mathrm{GND}(\Delta \wedge \neg\alpha, \{c_1, \ldots, c_k\})|_{N^{(i)}}$ is unsatisfiable **then**
            Increment $v$ and skip to the next $i$.
        **end if**
    **end for**
**end for**
Return $v/m$

---

**Theorem 15:** *Let $\delta, \gamma \in (0, 1)$ and $k \in \mathbb{N}$ be given. Suppose we have $m$ partial models drawn i.i.d. from a common distribution $D$ masked by a masking process $\Theta$, where $m \geq \frac{1}{2\gamma^2} \ln \frac{2}{\delta}$. (Here, $\ln$ denotes the natural logarithm.) With probability at least $1 - \delta$, Algorithm 1 returns a value $\hat{p}$ s.t.*

> **I** *if $\Delta \supset \alpha$ is at most $p$-valid, $\hat{p} \leq p + \gamma$*
>
> **II** *if there is a KB $\mathcal{I}$ such that*
>
> > *1. $\Delta \wedge \mathcal{I} \models \alpha$,*
> > *2. the rank of $\Delta \wedge \mathcal{I}$ is at most $k$, and*
> > *3. with probability at least $p$ over partial models $N \in \Theta(D)$, there exists names $c_1, \ldots, c_k$ not appearing in $\Delta$ or $\alpha$, such that every formula in $\mathcal{I}$ is witnessed true in $N$ for $c_1, \ldots, c_k$ together with the names appearing in $\Delta$ and $\alpha$*
>
> *then $\hat{p} \geq p - \gamma$.*

**Proof:** **Part I: $\hat{p} \leq p + \gamma$ if $\Delta \supset \alpha$ is at most $p$-valid.** We first note that when $\mathrm{GND}(\Delta \wedge \neg\alpha, C)|_{N^{(i)}} \models \bot$ for any set of names $C$, since $N^{(i)}$ is consistent with the actual model $M^{(i)}$ that produced it, $\mathrm{GND}(\Delta \wedge \neg\alpha, C)|_{M^{(i)}} \models \bot$ as well. Thus, in this case, $\mathrm{GND}(\Delta \wedge \neg\alpha, C)$ is falsified by $M^{(i)}$. Since $|C|$ is at least the rank of $\Delta$, it is easy to see that $\mathrm{GND}(\Delta \wedge \neg\alpha)$, which is logically equivalent to $\Delta \wedge \neg\alpha$, is falsifiable at $M^{(i)}$. So, it must be that the negation of that theory (i.e., $\Delta \supset \alpha$) is satisfied at $M^{(i)}$.

Now, $\Delta \supset \alpha$ is by definition $p$-valid with respect to this distribution on $M^{(i)}$ if the probability that $\Delta \supset \alpha$ is satisfied by each $M^{(i)}$ is $p$. Moreover, it follows immediately from Hoeffding's inequality that for $m \geq \frac{1}{2\gamma^2} \ln \frac{2}{\delta}$, the probability that the fraction of times $\Delta \supset \alpha$ is satisfied by $M^{(i)}$ (out of $m$) exceeds $p$ by more than $\gamma$ is at most $\delta/2$. Thus, $\hat{p}$, which is at most the fraction of times $\Delta \supset \alpha$ is actually satisfied by $M^{(i)}$, likewise is at most $p + \gamma$ with probability at least $1 - \delta/2$.

**Part II: rate of witnessing an implicit KB lower bounds $\hat{p}$.** Note that by the grounding trick (Theorem 2), $\Delta \wedge \mathcal{I} \models \alpha$ implies that for any set of names $c_1, \ldots, c_k$ not appearing in $\Delta$ or $\alpha$, $\mathrm{GND}(\Delta \wedge \mathcal{I} \wedge \alpha, \{c_1, \ldots, c_k\}) \models \bot$. Suppose that $\mathcal{I}$ is witnessed true for $c_1, \ldots, c_k$ together with the names in $\Delta$ and $\alpha$ in $N^{(i)}$. We note that in the restricted formula $\mathrm{GND}(\Delta \wedge \mathcal{I} \wedge \neg\alpha, \{c_1, \ldots, c_k\})|_{N^{(i)}}$, the groundings of formulas in $\mathcal{I}$ all simplify to 1 (true), and so $\mathrm{GND}(\Delta \wedge \mathcal{I} \wedge \neg\alpha, \{c_1, \ldots, c_k\})|_{N^{(i)}} = \mathrm{GND}(\Delta \wedge \neg\alpha, \{c_1, \ldots, c_k\})|_{N^{(i)}}$. Thus, $\mathrm{GND}(\Delta \wedge \neg\alpha, \{c_1, \ldots, c_k\})|_{N^{(i)}} \models \bot$, so $v$ is incremented on this iteration. Thus, indeed, $\hat{p} = v/m$ is at least the fraction of times out of $m$ that $\mathcal{I}$ is witnessed true for some set of $k$ names. It again follows from Hoeffding's inequality that for $m \geq \frac{1}{2\gamma^2} \ln \frac{2}{\delta}$, this is at least $p - \gamma$ with probability $1 - \delta/2$.

By a union bound, the two parts hold simultaneously with probability at least $1 - \delta$, as needed. ∎

In essence, the no-overestimation condition is a *soundness* guarantee and the no-underestimation condition is a limited *completeness* guarantee: in other words, if the query logically follows from the explicit KB and examples then the algorithm returns success with an appropriate $\hat{p}$, and vice versa. Note that the number of examples $m$ needed (to answer a single query) depends only on the desired accuracy $\gamma$ and confidence $\delta$. It is independent of the size of the KB, the number of predicates, etc.

**Example 16:** Continuing Examples 12 and 14, we noted that the $\forall$-clause

$$\forall x \neq logan \supset [Mutant(x) \supset Teammate(x,logan)]$$

was witnessed w.r.t. $\Delta$ for partial models such as {*Teammate(scott,logan)*, *Teammate(jean,logan)*} or {*Teammate(ororo,logan)*, *Teammate(kurt,logan)*}. This formula could serve as an implicit KB if $\Theta(D)$ produces such examples; it completes a proof of *Mutant(logan)* by first inferring *Mutant(x)* for some $x \neq logan$ from the first rule of $\Delta$, using this implicit KB formula to infer *Teammate(x,logan)*, and finally using the second rule of $\Delta$ to infer *Mutant(logan)*. In these partial models, respectively, the restricted grounding of $\Delta$ correspondingly produces *Mutant(scott)* and *Mutant(scott)* $\supset$ *Mutant(logan)*, or *Mutant(ororo)* and *Mutant(ororo)* $\supset$ *Mutant(logan)*, which in each case allows us to prove the query *Mutant(logan)*, via a different individual depending on the names mentioned in the partial example. Observe that $\Delta$ does not allow us to infer that the *Teammate* relation holds for any individuals, whereas the data alone, which *only* gives positive examples of the *Teammate* relation, is not adequate to infer the *Mutant* relation. We need *both* to establish *Mutant(logan)*.

## 6   Tractable Reasoning

Algorithm 1 reduces reasoning with implicit learning to deciding entailment. In order to obtain a tractable algorithm, we generally need to restrict the reasoning task somehow. One approach, taken in the previous work on propositional implicit learning [11], is to "promise" that the query is provable in some low-complexity fragment; for example, it is provable by a small treelike resolution proof (where "small" refers to the number of lines of the proof). Equivalently, we give up on completeness, and only seek completeness with respect to conclusions provable in low complexity in a given fragment. In general, then, one obtains a running time guarantee that is parameterized by the size of the proof of the query. We can take a similar approach here, by using an algorithm for deciding entailment that is efficient when parameterized in such terms. In general, what is needed is a fragment for which we can decide the existence of proofs efficiently, and that is "restriction-closed," meaning that for any partial model $N$, if we consider the restriction of each line of the proof, we obtain a proof in the same fragment. Most fragments we might consider, including specifically treelike or bounded-width resolution, are restriction-closed. (See [10] for details.)

We will motivate an entirely new strategy here, which offers a semantic perspective to the proof-theoretic view in [11]. One classically sound model-theoretic approach to constraining propositional reasoning is to limit the power of the reasoner, as represented, for example, by the work on tautological entailment [16]. More recently, [20] suggest a simple evaluation scheme for proper$^+$ KBs that gradually increases the power of the reasoner: level 0 is standard database lookup together with unit propagation, level 1 allows for one case split in a clause, level 2 allows two case splits, and so on. The formal intuition is as follows: suppose $s$ is a set of ground clauses and $\phi$ is a ground query, and let us say its a clause for simplicity. Let $\mathcal{U}(s)$ denote the the closure of $s$ under unit propagation, defined as the least set $s'$ satisfying: (a) $s \subseteq s'$ and (b) if literal $l \in s'$ and $(\neg l \vee c) \in s'$ then $c \in s'$. Then let $\mathcal{V}(s)$ define all possible weakenings: $\{c \mid c$ is a ground clause and there is a $c' \in \mathcal{U}(s)$ s.t. $c' \subseteq c\}$. Then we define $s \models_z \phi$ (read: "entails at levels $z$") iff one of the following holds:

- *subsume:* $z = 0$, and $\phi \in \mathcal{V}(s)$;
- *split:* $z > 0$ and there is some clause $c \in s$ such that for all literals $l \in c$, $s \cup \{l\} \models_{(z-1)} \phi$.

For small values of $z$, entailment at level $z$ is tractable to decide as well as sound:

**Theorem 17:** *[20] Suppose $\Delta, \phi$ are propositional formulas and $z \in \mathbb{N}$. Then, determining if $\Delta \models_z \phi$ can be done in time $O((|\phi||\Delta|)^{z+1})$. Moreover, if $\Delta \models_z \phi$ then $\Delta \models \phi$.*

We will now see how to leverage these results. First, however, we need the equivalent to restriction-closed, as discussed above.

**Proposition 18:** *Suppose $\phi, \Delta, z$ are as above. Then if $\Delta \models_z \phi$, and $N$ is any partial model then $(\Delta|_N) \models_z (\phi|_N)$.*

Basically, if $\phi$ is entailed at level $z$ from $\Delta$, then any restriction of $\phi$ under $N$ must also be entailed by $\Delta$ restricted to $N$, at least at level $z$ if not lower. Notice that restricting a ground formula is equivalent (w.r.t. satisfiability) to simply conjoining the literals true at $N$ with that formula, from which the proof follows. Now, recall from Theorem 2, given a proper$^+$ KB $\Delta$ and ground query $\phi$, we have $\Delta \models \phi$ iff $\mathrm{GND}^-(\Delta \wedge \neg\alpha)$ is unsatisfiable. Here, since $\alpha$ is already ground, we really only need to make sure that $\Delta$ is ground wrt all the names in $\Delta \wedge \neg\alpha$ and $k$ new ones, $k$ being the rank of $\Delta$. So let $\mathrm{GND}^\alpha(\Delta)$ denote precisely such a grounding of $\Delta$. It then follows that $\mathrm{GND}^\alpha(\Delta) \models \alpha$ iff $\Delta \models \alpha$. It is easy to show that the same holds for $\models_z$ as well [20]. So let Algorithm 1$'$ be exactly like Algorithm 1 except that it takes an additional parameter $z$ (for limited reasoning) and replaces the following check:

$\mathrm{GND}(\Delta \wedge \neg\alpha, \{c_1, \ldots, c_k\})|_{N^{(i)}}$ is unsatisfiable          **with**

$\mathrm{GND}(\Delta, \{c_1, \ldots, c_k, d_1, \ldots, d_m\})|_{N^{(i)}} \models_z (\alpha|_{N^{(i)}})$, where $\{d_1, \ldots, d_m\}$ is the set of names appearing in $\alpha$ but not in $\Delta$.

**Theorem 19:** *Let $\delta, \gamma \in (0, 1)$, $k \in \mathbb{N}$, $m \geq \frac{1}{2\gamma^2} \ln \frac{2}{\delta}$, and let $z \in \mathbb{N}$. Then with a probability at least $1 - \delta$, Algorithm 1$'$ returns a value $\hat{p}$ such that:* **(I)** *and* **(II)** *is as in Theorem 15 except for* **(II.1)** *which states that $\Delta \wedge \mathcal{I} \models_z \alpha$. The algorithm runs in time $O((|\alpha||\mathrm{GND}^\alpha(\Delta)|)^{z+1}m)$.*

**Discussion.**   Interestingly, in [21], it is shown that reasoning is also tractable in the first-order case if the knowledge base and the query both use a bounded number of variables. This would then mean that we would no longer be limited to ground queries and can handle queries with quantifiers. This direction is left for future research. Nonetheless, we note that deciding quantified (as opposed to ground) queries appears to demand more from learning. In general, in an infinite domain, we cannot hope to observe in a finite partial model that universally quantified formulas are ever true. Thus, we anticipate that extensions that handle queries with quantifiers will need a substantially different framework, presumably with stronger assumptions. One possible framework takes a more *credulous* approach to the learning problem (in contrast to our *skeptical* approach based on witnessing truth): we suppose that when a formula is frequently false on the distribution of examples, we also frequently obtain a partial model that witnesses the formula false—e.g., a partial model in which a binding of a candidate $\forall$-clause falsifies it. This is undoubtedly an assumption about the benevolent nature of the environment, captured as the notion of *concealment* in [25], but it does make learning conceptually simpler. In this framework, one permits all conclusions that are not explicitly falsified. Whether such an idea can be used for inductive generalization of FOL formulas over arbitrary distributions remains to be seen.

## 7   Conclusions

In this work, we presented new results on the problem of answering queries about formulas of first-order logic (FOL) based on background knowledge partially represented explicitly as other formulas, and partially represented as examples independently drawn from a fixed probability distribution. By appealing to the paradigm of implicit learnability, we sidestepped many major negative results, leading to a learning regime that works with a general and expressive FOL fragment. No restrictions were posed on clause length, predicate arity, and other similar technical devices seen in PAC results. Overall, we hope the simplicity of the framework is appealing to the readers and hope our results will renew interest in learnability for expressive languages with quantificational power.

**Acknowledgements**

V. Belle was supported by a Royal Society University Research Fellowship. B. Juba was supported by NSF Award CCF-1718380. This work was partially performed while B. Juba was visiting the Simons Institute for the Theory of Computing. We thank our reviewers for their helpful suggestions.

## Footnotes

[1]Our language $\mathcal{L}$ is essentially equivalent to standard FOL together with a unique-name assumption for infinitely many constants [17, Definition 3].

[2]Because equality is treated separately, atoms and clauses do not include equalities.

[3] Valiant [31] uses a fragment of FOL for which propositionalization is guaranteed to yield a small propositional formula, and only considers such a reduction to the propositional case.

[4]Note that since we assume that the resulting partial models are finite and thus countable, as long as the masking processes are measurable functions w.r.t. the joint probability measure, every event defined in terms of the partial models is a countable union of measurable events, and thus measurable.

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
