[Reviews · NeurIPS 2019]

Reviewer 1



Solid paper with interesting theoretical results. In a high level, the challenge the authors had to solve is: how to achieve or even state learning when there are so many sources of infinities, such as example models and elements in each model's underlying universe? Of course, the paper considers a specific instance of this general question that arises in the context of a restricted version of first-order logic and a particular class of models for the logic. But I found its formulation of the problem elegant and the solution nice. Specifically, I liked the idea of using the fact that formulas cannot distinguish certain non-explicitly named elements and cannot count in a sense more than certain syntactically determined numbers. Also, the use of partial model and random-masking process is quite clever.

Reviewer 2



This paper is generally well written and clear, albeit targeting readers with formal backgrounds. The quality of the paper seems high in terms of its formal claims. The proposed mechanism is remarkable simple, making this an attractive approach. I really like the idea behind not making learning explicit (as opposed to rule induction for example). I have three main concerns about this paper: - In general it is very close to Juba's 2012 work [1]. In that work, a propositional variant is proposed. The work here extends the approach. This is a sensible idea, and well executed as far as I can tell, but the delta is still relatively small and the change feels relatively mechanic. That's not a problem per se, but means that in terms of originality this paper isn't as convincing. The authors make it worse by using verbatim copies of some of the text in [1], such as the first paragraph of section 3. - It's not clear what the significance of this result is. The authors do go beyond propositional logic, but make little effort to motivate the specific fragment and tell us what we can now do we couldn't before. For example, could this be contrasted with explicit rule learning and could we empirically see when and how it improves over that. While I understand this a primarily theoretical contribution, I think for a venue like NeuRIPS the authors need to do a better job in positioning and motivating their work in the broader statistical relational learning context. - I found it difficult to get an intuition how "implicit learning" really operates. For example, if I see a bunch of examples where human(x) => mortal(x), but I don't have any such formula my background, how is this formula implicitly learnt? I think I understand the algorithm but I have no intuition how it achieves this. Good running examples could help. [1] https://arxiv.org/pdf/1209.0056.pdf (the authors do cite this paper and attribute it appropriately) Update: after author response and reviewer discussion I increased my score from 4 to 6. I do think this is an important and novel theoretical contribution. I would still encourage the authors to make paper more accessible in terms of exposition and contextualisation for the non PL/Logic folks in the community. I would also *strongly* encourage them to not copy and paste whole paragraphs from Juba 2012.

Reviewer 3



• I struggled to follow the paper, which in my view has several reasons - While I am familiar with statistical relational learning, I lack some of the background to digest the paper in full. The paper is not very self contained and many concepts are assumed to be known to the reader or only touched upon briefly. For example, what is a universal closure (L100), - The paper is very dense, not leaving much space for intuitive explanations or examples. My impression is that this work might better fit into a journal format. I could also imagine it would make sense to move most of the proofs to the appendix and use the space to add examples and more intuitive explanations. A good example of what I mean can be found in lines 214 to 221. The explanation there was really helpful and I would want to see more of these throughout the paper. - The notation, while I assume common in this subfield, clashes with symbols commonly used in machine learning (e.g. θ,Δ,α) - The paper is notation heavy. I understand it is a theoretical paper and space is limited, but I felt at various points the authors should have spent more time spelling things out (e.g. In Theorem 16 it would literally have been shorter to briefly restate what δ,γ,k,m are rather than referring to Theorem 13). • How does learning from entailment here is connected to learning from entailment in the inductive logic programming literature (e.g. Stephen Muggleton. Inductive logic programming. New Generation Comput., 8(4): 295–318, 1991.)? • Weaknesses of the proposed approach are not discussed. For instance, is the time complexity mentioned in Theorem 14 good or bad? To what extent does it have implications for practical application (e.g. Knowledge Bases with thousands, tens of thousands, hundreds of thousands of facts)? My main problem is gauging the significance of the presented work. I think this is in part me not able to understand the paper in detail, but also due to the fact that there is no empirical validation of the presented approach. Minor comments • L65: I would add neuro-symbolic learning to the unification of statistical and logical representation here. See for example - Cohen, William W. "Tensorlog: A differentiable deductive database." arXiv preprint arXiv:1605.06523 (2016). - Rocktäschel, Tim, and Sebastian Riedel. "End-to-end differentiable proving." Advances in Neural Information Processing Systems. 2017. - Evans, Richard, and Edward Grefenstette. "Learning explanatory rules from noisy data." Journal of Artificial Intelligence Research 61 (2018): 1-64. - Manhaeve, Robin, et al. "Deepproblog: Neural probabilistic logic programming." Advances in Neural Information Processing Systems. 2018. • L81: I am not familiar with the last logical connective. • L107: What does "maximal" mean here? • L111: What does "eθ" refer to? Applying substitutions θ to e? • L116: I am confused since as far as I understand z would be a rank and a name here? • L155-7: I think this is a good summary of the goal of the paper and should be stated earlier. • L175: Can you elaborate why summing εi is valid here? My intuition is that this only works if the clauses are independent. Maybe this is trivially the case, but it is not obvious to me. • L220: This seems to depend on the distribution of test queries though. Are you assuming the follow the same/similar distribution as test queries? UPDATE: After considering the rebuttal by the authors and discussions with the other reviewers I am happy to increase my score. That said, I strongly encourage the authors to make the paper more accessible and improve its clarity. I particularly believe more running examples and intuitive explanations would make the paper stronger.

[Author Response · NeurIPS 2019]

**Reviewer 1**   Sorry for the confusion. The argument is roughly that because Theorem 4 establishes that $\mathrm{GND}(\alpha, 1)$ is satisfiable if and only if $\alpha$ is in the countably infinite domain, and Lemma 5 establishes that the introduction of extra names in $\mathrm{GND}(\alpha, z)$ preserves satisfiability, we obtain satisfiability under the larger, common subset of names used in $\mathrm{GND}^-$ as well. Please see the Appendix of Ref. 1 for a formal proof.

For your second question, note that for our purposes, it is enough to only consider masking functions that mask all but a finite subset of the domain, thus producing finite-size partial models. Thus, we can take both our masking function and the "application function" $\mathrm{app}(M, \Theta)$ to have a countable range, not something with continuum (or larger) cardinality. In particular, then, we get measurability: these discrete output sets can be defined by a countable union over the resulting finite partial examples. The preimage of a single partial model in turn will be a measurable set for $M$ and $\Theta$, given that $M$ is a measurable function: by definition, the preimage for a measurable $M$ of that partial model is a measurable set.

**Reviewer 2**   The representation lanaguage $\mathrm{proper}^+$ that we use is emerging as a popular representation language. FOL with universal quantifiers is widely used to express inductive properties in mathematics but also to represent social networks and graphs. For computability results, usually the finite domain assumption is made, but interestingly, $\mathrm{proper}^+$ seems to allow us to go beyond the closed-world assumption. (And unlike description logics, arity restrictions are also not needed.) We note that beyond the fact that $\mathrm{proper}^+$ extends incomplete databases (L104–107), for example (Liu and Lakemeyer, 2009) show how to represent a certain family of "local" action models for planning within the fragment of $\mathrm{proper}^+$ for which Theorems 14/16 give polynomial-time reasoning. There is also a variant for epistemic planning (Muise et al. 2015) where one reasons about the mental states of other agents, and we expect analogous extensions of our work may contribute to that direction too. In particular, our approach applies to infinite domains, or even simply large domains without resorting to directly considering all groundings of the atomic formulas, in contrast to Juba's work. Note that even in moderate size, finite domains, the number of groundings grows exponentially with the arity of the formulas under consideration, and thus quickly grows infeasible to represent as a propositional formula (which is required for Juba's approach).

Implicit learning works because the partial models themselves compactly encode all of the rules that could be learned from those models. So instead of trying to learn a large set of rules from the models and hoping that these rules will permit us to derive the desired conclusions, we use the models directly to answer queries.

**Reviewer 3**   Sorry for the terse exposition. The *universal closure* is the result of placing a universal quantifier on each free variable appearing in the formula. $\supset$ (L81) denotes implication. Maximality (L107) refers to the database case, in which every true literal is included; by contrast we have a set of clauses that may not specify all of the true literals. $e\theta$ (L111) indeed refers to applying the substitution $\theta$ to $e$. $z$ (L116) is the rank, which yes is an integer. We use $\mathbb{N}$ as the set of names for convenience, but it is not important here that $z$ could be interpreted as a name. In Proposition 8 (L175) we simply take a union bound over the error events which have respective probabilities $\epsilon_i$. Note that $1 - \epsilon'$ validity only requires that the total probability of the error events is *at most $\epsilon'$*. The union bound applies to any set of events and in particular does not require independence. The actual guarantees of the informal discussion on L220 are formalized in Theorem 13. There is not a requirement on a distribution of queries. Rather, what we promise is that I: we will not (significantly) overestimate the validity of a query and II: we guarantee that our estimate of the validity is (approximately) at least the probability that some suitable implicit KB $\mathcal{I}$ is witnessed.

Learning from entailment is pretty different from what we seek here: it asks us to produce a set of formulas $H$ that replicates a desired set of entailment judgments, e.g., that $\phi_1$ is entailed but $\phi_2$ is not, etc. Our task formulation is much closer to learning from interpretations in ILP, where our partial models are partial intepretations. In that task, one is given a set of background knowledge formulas $B$ and a set of models $x_1, x_2, \ldots$ and seeks an additional set of background knowledge $H$ such that $B \wedge H$ is consistent with the given models. One could subsequently answer entailment queries against $B \wedge H$. The difference is first that ILP only seeks an $H$ that is consistent with the examples, and does not seek to analyze the degree to which the resulting formulas capture an unknown, ground-truth process that produced the example models $x_1, x_2, \ldots$. In particular, there is no sense in which the resulting judgments $B \wedge H \models \alpha$ are "correct" or "incorrect" in ILP, unless we use a "closed-world" assumption or something similar, that leads the set of models to *define* a single "correct" $H$. Even so, in practice, ILP often requires significant restrictions on the set of clauses permitted in $H$ to ensure that there is a finite Herbrand base of atoms to search through. We will include a discussion on this in the paper.

The time complexity bound is good in the sense that for fixed approximation and confidence parameters $\gamma$ and $\delta$, the time complexity of querying the implicit KB is equivalent to a constant number of queries for an explicit KB (cf. Theorem 14). So it remains tractable.

**References** (a) Y. Liu and G. Lakemeyer. On first-order definability and computability of progression for local-effect actions and beyond. In Proc. IJCAI, pages 860–866, 2009. (b) C. J. Muise, V. Belle, P. Felli, S. A. McIlraith, T. Miller, A. R. Pearce, and L. Sonenberg. Planning over multi-agent epistemic states: A classical planning approach. In AAAI, 2015.

[Meta-Review · NeurIPS 2019]

After fairly in-depth discussion, the reviewer consensus for this paper has shifted towards acceptance. The reviewers agree that the paper presents a simple but general framework for implicitly learning to reason under some constraints. The paper makes an important theoretical contribution, of a type which is welcome but perhaps too rarely is included in NeurIPS and similar conferences, and should be of interest to the community. The reviewers, however, all flagged that the paper could use some fairly significant improvement when it comes to presentation and clarity, and that it would benefit from clear exposition of the formalization section, and running/motivating examples to serve as an intuition pump for those readers with a less formal background. Overall, acceptable, but the authors are *strongly* encouraged to take heed of the recommendations of all three reviewers when preparing their camera ready, should the paper be accepted by the PC.